# HESS Opinions: Reflecting and acting on the social aspects of modeling

Janneke O.E. Remmers<sup>1,\*</sup>, Rozemarijn ter Horst<sup>2</sup>, Ehsan Nabavi<sup>3</sup>, Ulrike Proske<sup>1</sup>, Adriaan J. Teuling<sup>1</sup>, Jeroen Vos<sup>2</sup>, and Lieke A. Melsen<sup>1,\*</sup>

Abstract. Hydrological models are generally acknowledged as subjective and uncertain, yet they are often still perceived as neutral, meaning they are seen as not taking sides. This notion of neutrality has several, potentially harmful, consequences. One is the marginalization of certain stakeholders: failing to acknowledge or incorporate alternative perspectives on the issue, which might have warranted a different (modeling) approach. In the critical social sciences, the non-neutrality in methods and research results is an established topic of debate. Thus we propose that in order to deal with it in hydrological modeling, the hydrological modeling network (from commissioner to modeler to end-user) can learn from, and with, critical social sciences. This is a call for responsible modeling – modeling that is accountable, transparent, power-sensitive, situated and reproducible and this responsibility is carried by all actors related to the modeling study. To support our proposition, we structure our argument around four key pillars: (1) the social dimensions of and within hydrological modeling, (2) insights from the critical social sciences, (3) building bridges between disciplines, and (4) reflecting on what the hydrological modeling network can learn. The main take-away, from our perspective, is that responsible modeling is a collective responsibility, shared by all actors in the modeling network. We provide several actionable recommendations for individual actors to increase their share in facilitating responsible modeling.

<sup>&</sup>lt;sup>1</sup>Hydrology and Environmental Hydraulics Group, Wageningen University, Wageningen, The Netherlands

<sup>&</sup>lt;sup>2</sup>Water Resources Management Group, Wageningen University, Wageningen, The Netherlands

<sup>&</sup>lt;sup>3</sup>Australian National Centre for the Public Awareness of Science, Australian National University, Canberra, Australia

<sup>\*</sup>Corresponding author: Lieke A. Melsen (lieke.melsen@wur.nl)

### 1 Introduction

40

Models are frequently used tools, both to support decision-making - for example, during drought situations as discussed in Lam et al. (2025) - and for scientific research, as illustrated by the bibliometric analysis in Burt and McDonnell (2015). In both cases, the modeling process involves choices and interpretations (Refsgaard, 1996; Savenije, 2009), raising important questions about subjectivity, transparency, and the narratives models help construct - especially given their wide application. In this context, we understand hydrological modeling as a practice that encompasses the full process from developing and implementing model code to setting up and applying the model to address specific question or issue. Our discussion in this paper is primarily informed by experiences with numerical models; however, we argue that the principles and concerns we raise also apply to data-driven modeling approaches.

Although model developers and users generally acknowledge that models are hypotheses (Savenije, 2009), determined by experts' system understanding and subject to subjectivity and uncertainty (McMillan et al., 2023), this is rarely connected to deeper reflections on how this shapes certain narratives that benefit some while disadvantaging others: The model is recognized as (partly) subjective, yet still perceived as neutral that is, as remaining impartial or not taking sides. This perception is shared not only by the modelers themselves, but certainly also by commissioners and other end-users. Similarly, it is generally acknowledged that models influence society - for example, by supporting decision-making during events such as the COVID-19 pandemic (Nabavi, 2022; Saltelli et al., 2020). At the same time, this notion of model neutrality presumes that the model itself is not influenced by society, and that the model provides unbiased information. However, we argue that hydrological modeling takes place within a socio-political context, which affects what the model can do, and for whom (Krueger et al., 2012; Mayer et al., 2017; Wesselink et al., 2017; Melsen et al., 2018; Packett et al., 2020, visualised in Fig. 1). Models are shaped and influenced by social and political dynamics (both at societal level and within the modeling community) and, in turn, influence them - for example by informing policies or infrastructure designs that may benefit certain groups or ecosystems while disadvantaging others.

Perceiving modeling as neutral has several, potentially harmful, consequences. Neutrality implies that all people and aspects are treated equally. This is not the case (Doorn, 2017; Packett et al., 2020). For example, models are always simplifications of reality, and therefore choices are made on what to represent in the

**Figure 1.** General overview of the social context of models. The societal context (blue circles) are worked out in Argument 1 of Section 3. The modeling community context (orange circles) are elaborated in Argument 2 of Section 3. The societal and ethical consequences of models are described in Argument 3 of Section 3.

model, what not, and how (Frigg and Hartmann, 2024; Refsgaard, 1996; Savenije, 2009). As a result, the unrepresented processes and aspects are marginalized and become invisible. This can result in injustices: some groups being overlooked, some interest being prioritized, or some ways of understanding sidelined (Doorn, 2017; Zwarteveen and Boelens, 2017), obfuscated by assumed neutrality.

Simultaneously, ignoring the political side of models, meaning how power plays a role, may impede their potential or effectiveness (Beven et al., 2022; ter Horst et al., 2023; Saltelli and Di Fiore, 2023): representing certain processes or not may facilitate the interests of powerful stakeholders. For instance, Kroepsch (2018) describes a case in which a groundwater-extracting industry had a vested interest in excluding surface water–groundwater interactions from a model, in order to avoid the obligation of compensating surface water rights holders. Another example, at the scientific community level, is that some large institutions fund hydrological research with the requirement to use their data (Melsen, 2022). This highlights their position of power, as it leads to scientific publications that use, and thereby legitimize, their data, even when better

alternatives may have been available. Acknowledging the political side of modeling can help better connect models to the specific needs within the problem being addressed, and understand the context in which the model was developed.

In the critical social sciences – the sciences dealing with critical questions of power relations, especially oppression and domination (Watts and Hodgson, 2019) – the non-neutrality in methods and research results has been a topic of debate for a longer period already (Mendelsohn, 1977; Latour, 1990; Law, 2004; Sismondo, 2011). Different disciplines within the critical social sciences, such as Science and Technology Studies (STS) and political ecology, provide insights into how to analyze and deal with non-neutrality. We believe that the hydrological modeling community can benefit from engaging with critical social sciences, both to learn from them and to collaboratively advance our understanding of the role of models.

Research on infrastructures within STS offers a clear example of how design is not neutral. A well-known example is the study of Star and Strauss (1999), who examined the everyday work of hospital nurses. Through ethnographic observation, they revealed how standardized forms and technologies often failed to capture the complexities of nursing practices. Nurses developed informal workarounds to fit this complex reality into the official documentation; the official documentation typically reflected the perspectives of doctors while sidelining the experiential knowledge of nurses. As such, this case underscores how standards and technologies, in this case digital documentation, tend to represent certain values and perspectives over others.

The field of political ecology studies the role of power (aka, politics) and the broader political context of environmental issues. An example is the study on soil erosion in Nepal (Blaikie, 1985). Soil erosion was often framed as a result of poor farming practices by local farmers. Blaikie demonstrates how this is also the result of power structures: Since the majority of land was held by a small elite, small local farmers relied on tenant land farming. Because these leases could be terminated at any time, tenant farmers had little incentive to invest in long-term, sustainable practices like erosion control. This suggests that solutions should focus not merely on training farmers in improved practices, but on enhancing livelihood security.

These examples illustrate how both STS and political ecology provide broader perspectives on the use of technology and the framing of environmental issues. Such a broader perspective, accounting for whose perspectives were involved in the technical design, and evaluating broader political contexts, can enrich our understanding of hydrological models and their place in society. For example, they can reveal how models

may be designed with specific viewpoints in mind, potentially marginalizing alternative perspectives, or how models are employed to address problems that appear technical at first but are, in fact, deeply socio-political.

Similar dynamics can be identified for hydrological models. For example, models used in flood studies may overlook informal settlements located in floodplains, thereby marginalizing the people who live there (Wesselink et al., 2017). This brings forward the non-neutrality of hydrological models; what is represented and how, matters. In order to take this into account, we propose that the hydrological modeling network, which we define as all actors, i.e. funders, commissioner, modelers, users, decision-makers, involved in and influencing the modeling study, can learn from, and with, critical social sciences. This is a call for responsible modeling – modeling that is accountable, transparent, reproducible, power-sensitive, situated, and inclusive of diverse knowledges and interests – and this responsibility is carried by all actors related to the modeling study.

95

110

We are aware that our argument is not new and has been brought up in different terms and ways across the hydrological modeling network. Part of this comes from our own contributions to this debate (ter Horst et al., 2024; Melsen, 2022; Nabavi, 2022; Remmers et al., 2024; Alba et al., 2025a), but we also acknowledge active research communities in Australia working on good modeling practices and model governance (Hamilton et al., 2022; Jakeman et al., 2006, 2024), work done in Germany on situated modeling (Klein et al., 2024; Krueger et al., 2012; Krueger and Alba, 2022; Alba et al., 2025b), ongoing research in France (Molle, 2009; Venot et al., 2014), Post-Normal science (Funtowicz and Ravetz, 1993; Petersen et al., 2011; van der Sluijs, 2002) and sensitivity auditing (e.g. Puy et al., 2023; Saltelli and Di Fiore, 2023), work done in the Chesapeake bay (Deitrick et al., 2021; Lim et al., 2023), and the Open modeling Foundation initiative (OMF, SA). While being far from complete, this list shows that different research groups actively contribute to this topic. That being said, from experience we know that the effects of these studies are often limited in practice, and therefore we provide here a clear overview of arguments to invite the hydrological (modeling) community to join the conversation on the non-neutrality of models, as well as to engage in a constructive way.

To support our proposition, we structure our argument around four key pillars: (1) the social dimensions of and within hydrological modeling, (2) insights from the critical social sciences, (3) building bridges between disciplines, and (4) reflecting on the lessons the hydrological modeling network can draw. Within each pillar, we present sub-arguments that support our central claim: that the hydrological modeling network can learn

from, and with, the critical social sciences to better understand the role of hydrological modeling in society.

First, we elaborate on our own positionality in this debate.

### 2 Positionality of the authors

140

To promote transparency and encourage reflection, we begin by outlining our backgrounds and our reasons for engaging with this topic through a positionality statement, further discussed in Section 4.

We are a group of scholars who critically engage with the practice of modeling from a range of disciplinary and personal perspectives. Our academic backgrounds span hydrological and climate modeling, water governance, science and technology studies (STS), and political ecology. Some of us work directly with models, while others approach modeling as an object of critique. This range of experience brings together both insider and outsider perspectives within the hydrological modeling community - the intended audience of this piece. As such, we are able both to speak in the language of the hydrological modeling community, and to question some of its internalized assumptions and standards. This positioning has influenced the arguments we present and how we construct them - often beginning from the model itself.

Our perspectives on modeling range from pragmatic to deeply skeptical. Some of us actively use models in our work, while others grapple with finding ways to use models while also acknowledging their limitations, partiality and inherent injustices. For some of us, the entry point into this conversation was methodological, arising from concerns about uncertainty and limitations in model design, while for others it was rooted in confronting structural inequalities that can be reinforced through modeling practices.

Most of us are affiliated with institutions in the Global North and hold relatively privileged positions within academia. This affects how we access, use, and critique modeling tools. We recognize that our academic and geographic positions may limit our ability to fully engage with those most affected by the outcomes of modeling processes. While some among us have close relationships with communities that have been marginalized through models, we acknowledge that the perspectives presented here are still shaped primarily by voices of privilege - while one of our core arguments is to give voice to silenced groups.

Several of us are involved in teaching hydrological modeling. Some have already begun integrating reflexive practices into their teaching, through discussions of ontology, uncertainty, and situated knowledge, though we recognize that this work is still evolving. The diversity of backgrounds and experiences among

us has been a source of productive dialogue, particularly in shaping the framing and language of this piece. Despite our differences, we found a strong sense of shared concern and general agreement about the need for deeper reflexivity in modeling practices.

# 3 Social aspects in hydrological modeling

165

The first pillar supporting our proposition concerns the social aspects already present in and around hydrological modeling. Demonstrating this highlights the importance for the hydrological modeling network to acknowledge that modeling is not a neutral activity, but actively shapes worlds (Krueger and Alba, 2022). This pillar is underpinned by three arguments.

First, the problems hydrological modelers study are **embedded within society, with all its social processes** (Arg. 1, blue circles in Fig. 1). Water availability in rivers is impacted by land use changes (Teuling et al., 2019; Wamucii et al., 2021); unsustainable management of groundwater abstraction has social and political consequences (Nabavi, 2018; Sanz et al., 2019); sea level rise necessitates societies to adapt to the risks it brings (Irani et al., 2024; Kopp et al., 2019). The awareness of the entanglement of hydrology with society led to the initiation of the field of socio-hydrology or hydro-sociology (Sivapalan et al., 2012; Krueger et al., 2016; Melsen et al., 2018; Ross and Chang, 2020). These disciplines explore hydrological problems as integrated parts of society, often using stakeholder participation as an approach to include the different perspectives to an hydrological problem (ter Horst et al., 2024; Xu et al., 2018). That being said, it should be recognized that not only the challenges addressed with models are embedded in society, but that the modeling itself is also the result of the society in which it was shaped (Melsen et al., 2018; Riaux et al., 2023). Norms, values, and discourses commonly accepted within a society provide the space within which the hydrological model is developed and accepted. Even more, what is considered a problem is determined by societal standards held by the model commissioners, modelers and model-users. For instance, flood risk might be considered differently at different places.

Second, the **modeling process itself is a social product** (Arg. 2, orange circles in Fig. 1). This became already clear from Arg. 1 where we discussed how generally accepted norms and values are embedded in the framing of the problem and the model, but this is further emphasized by the modeling process, at a more technically detailed level, being dependent on dynamics in the modeling community. Modeling inherently

involves decisions underdetermined by empirical data and driven by social processes. Underdetermined decisions arise from equifinality, meaning that several options are not distinguishable from each other based on empirics, and as such are not 'objectively better' compared to each other (Beven and Freer, 2001; Butts et al., 2004; Ward, 2021; Winsberg, 2012). Although equifinality is often explored in the domain of parameter uncertainty, it can be extended to equifinality in methods or approaches, which might still produce different results or conclusions. Khatami et al. (2019), for instance, identified six facets of model equifinality, namely, model structure, parameters, performance metrics, initial and boundary conditions, inputs, and internal fluxes. As a result of equifinality, many underdetermined modeling decisions are now guided by social processes rather than epistemic or empirical criteria, introducing subjectivity and inter-modeler-variability (Remmers et al., 2024). These social processes include habit (Babel et al., 2019), institutional legacy (Addor and Melsen, 2019), and peer experience (Melsen, 2022). As elaborated in Melsen et al. (2025) for the Nash-Sutcliffe Efficiency, modeling standards are not purely technical but socially negotiated, in this example shaped by American engineering societies and an active modeling community that recommended and subsequently reinforced the use of the Nash-Sutcliffe Efficiency. Additionally, choices made early on in the modeling process can influence choices later on, creating so-called path dependency (Lahtinen et al., 2017; Lenhard and Winsberg, 2010). For example, the chosen model software limits the possible model settings (Remmers et al., 2024). Furthermore, Lane (2014) argues that the hydrological modeler is not separated from society, and thus is not separated from the problem they study - which links Arg. 1 and 2. Together, these studies highlight that modeling is not just a technical exercise, but a socially learned and negotiated practice.

180

Third, recognizing that models are shaped both by broader societal contexts (Arg. 1) and by the social dynamics within the scientific community (Arg. 2) is crucial, because models, in turn, shape society: **they have political and ethical implications** (Arg. 3, Fig. 1). Certain groups are included in, and benefit from, the modeling process, while others are excluded or disadvantaged (Beck and Krueger, 2016). The assumptions and decisions embedded in models reflect particular perspectives on reality (Nabavi, 2022; Saltelli and Di Fiore, 2023), and selecting one perspective inherently means sidelining others. This can result in social and environmental injustices (Thaler, 2021; Zwarteveen and Boelens, 2017).

Examples of how certain perspectives might be prioritized in model development are provided by Packett et al. (2020) along the lines of gender. They cite a case studied by Zwarteveen (2017) in Nepal, where men and women worked cooperatively as co-farmers but prioritized different aspects of water flow. Men,

responsible for land preparation, focused on water arriving at the start of the irrigation season, while women, who managed weeds, needed consistent water throughout the season. An irrigation distribution model optimized for either water arrival or water sustainment would thus benefit either men or women in their activities.

Nabavi (2025) presents a case that illustrates the broader socio-political context of modeling. In this instance, a hydrological model was employed to justify an interbasin water transfer to the historically significant city of Isfahan, Iran. The transfer was underpinned by a century-old narrative, with the model serving primarily to reinforce this story, framing upstream water as "lost" to the Persian Gulf unless redirected to Isfahan. In response, upstream communities developed a counter-model that accounted for ecological impacts and the livelihoods of upstream populations. Within this alternative framing which also emphasized upstream effects, the justification for the water transfer no longer held.

Stakeholder engagement can help bring these marginalized perspectives back into the modeling process (Packett et al., 2020; Xu et al., 2018), although such engagement also presents its own set of challenges (e.g., Reed et al., 2009; Turnhout et al., 2020). Considering the political and ethical dimensions of modeling is thus essential to foster more responsible modeling.

# 4 Insights from critical social sciences

Critical social sciences provide the theoretical frameworks and tools that can address the social aspects of hydrological modeling. Here, we will highlight three.

First, the critical social sciences have the **vocabulary to express the social aspects in hydrological modeling** (Arg. 4). This vocabulary is not (yet) common in the hydrological modeling network, where similar concepts are addressed more elaborately. For example, when we just described that 'the assumptions and decisions embedded in models reflect particular perspectives on reality' in the previous section (in Arg. 3), we could have also used the term 'situated', stemming from feminist theories (Haraway, 2013). This means that perspectives, such as represented with models, are shaped by social, cultural, historical, and geographical background (see also Klein et al., 2024; Alba et al., 2025b). Another example is the term 'ontology', meaning the study of the nature of things (Frigg and Hartmann, 2024; Wesselink et al., 2017). The way a modeler understands the world, will affect how they represent it. For example, hydrologists often distinguish between epistemic and aleatoric uncertainty (e.g. Beven, 2016). Recognizing the existence of aleatoric un-

certainty, that is, uncertainty due to inherent randomness in natural processes, presupposes a belief that the world is not entirely deterministic (otherwise it would have been epistemic uncertainty). This illustrates how one's worldview, or ontology, influences which types of uncertainty are considered meaningful to study. The same applies to 'epistemology', the theory of how we know what we know. Modeling aligns well with a Newtonian perspective, which assumes that natural laws can be discovered and represented objectively. In contrast, a constructivist would argue that all knowledge is socially constructed, and thus would immediately question the idea of a single 'best' model, highlighting the partial and situated nature of modeling. Knowledge of this vocabulary can enhance our understanding of and facilitate our discussion of the social aspects in hydrological modeling (Laplane et al., 2019). A good starting point to become acquainted with this terminology is Moon and Blackman (2014) for general terminology, followed by Wesselink et al. (2017) and Klein et al. (2024) for application within the field of hydrology and modeling.

Second, critical social scientists aim to reflect on their positionality and practice active reflexivity in their research (Arg. 5). A positionality is written to indicate a researcher reflects on their own relation to the subject of study (Lin, 2015; Njeri, 2021; Soedirgo and Glas, 2020). For example, critical social science disciplines using ethnographic methods – observing subjects in their own environment – often include a positionality, since the scientist's background influences the observations and interpretations they make. Hydrological modelers also have a personal perspective, or position (from Arg. 2), towards their subject through their own previous experience or the institute they work at or even their own personal interests and hobbies (Deitrick et al., 2021; Melsen, 2022; Packett et al., 2020). Modelers tend to make decisions based on these experiences or contextual factors (Krueger et al., 2012; Melsen, 2022; Remmers et al., 2024; Sanz et al., 2019). Reflecting on and being transparent about positionality can create more transparency regarding this personal context and assumptions made (Blackett et al., 2024; Klein et al., 2024; Wesselink et al., 2017). For example, Melsen (2022) includes a brief positionality for the interview study she did, highlighting how her own background has influenced the conducted interviews. Besides writing a positionality, active reflexivity - continual questioning of your own assumptions and biases - should also be done throughout the modeling process (Soedirgo and Glas, 2020). This entails documenting assumptions, normalising reflexivity, engaging others in the reflexivity, and publishing the modeler's reflexivity alongside the research. We acknowledge that publishing reflexivity through a positionality means being vulnerable and open. We believe this to be a strength, however, because the vulnerability and transparency can build trust in how models are used.

Additionally, it can inspire others to also reflect on or to become more open about their modeling practices and assumptions. As more people start to do this, it could change practices in the whole modeling network.

As a starting point, we also included our positionality in this paper (Section 2). To stimulate reflexivity and think about positionality, we refer to Holmes and Gary (2020). Also the overview in terminology provided in Malterud (2001) can be a useful resource.

Third, while clear terminology and increased reflexivity are valuable starting points, **basic understanding** of critical social sciences is needed to situate research in a broader context, to understand the possible positive and negative consequences of modeling, and to be able to identify who to empower and how (Arg. 6). This context is necessary, since hydrological modeling is both influenced by and contributes to the shaping of societal issues (from Arg. 1), the hydrological modeling process is a social product (from Arg. 2), and model results have political and ethical implications (from Arg. 3). The necessary basic knowledge should entail knowledge to place modeling results in the societal context (from Arg. 1) and reflect on potential ethical consequences of the results (from Arg. 3). Understanding of certain concepts of critical social sciences can also ease reflecting on the subjectivity in modeling (form Arg. 2). For instance, the vocabulary (from Arg. 4) can help initiating reflexivity. Ontology – studying the nature of things – can spark debate on the different perspectives people have of a hydrological system (Agrawal et al., 2024). Given the ethical implications of models, it is essential that modelers develop a sensitivity, or "antennae", for the political and ethical consequences of their work, something a basic understanding of critical social science can meaningfully support.

Recently, ethics of Artificial Intelligence has gained traction (Doorn, 2021; Maier et al., 2024; Nabavi et al., 2024), and rightly so. Interestingly, however, a comparable ethical movement has yet to emerge within the field of numerical modeling, despite the fact that many of the same critiques are applicable. As such, ongoing ethical discussions in AI can provide valuable guidance for developing an ethics of numerical modeling. For instance, Nabavi and Browne (2023) propose the Five Ps framework to guide AI researchers and practitioners to situate their modeling work as interventions within competing perspectives on what constitutes a problem and how that framing influences the kind of solutions considered. This Problem-solution dynamic can be mapped onto specific zones of intervention - Parameter, Process, Pathway, and Purpose - each representing a distinct leverage point with varying potential for change. For example, addressing responsible AI challenges within the "Parameter" zone often involves quantifiable refinements, such as numerical adjust-

ments or parameter tuning. In contrast, interventions in the Purpose zone engage with foundational questions concerning the values, norms, and worldviews embedded in modeling practices. These efforts prompt deeper reflection, such as: What broader societal or ecological goals, like equity or resilience, should guide modeling practices? This framework supports hydrological modelers in openly reflecting on their role in problem framing and discussing intervention zones. This framework, developed with Artificial Intelligence applications in mind, can directly be translated to the ethics of numerical (hydrological) modeling.

## 5 Building bridges between (two) scientific disciplines

Different researchers have been trying to build bridges between (the social and hydrological) disciplines (Krueger et al., 2016; Pulkkinen et al., 2022; Rödder et al., 2020; Ross and Chang, 2020; Venot et al., 2022; Zwarteveen and Boelens, 2017), but most remain within their own discipline. Hierarchy of sciences – the idea that certain sciences, such as physics, have a higher degree of consensus and scientific advancement than others, such as social sciences – reinforces this way of thinking and acting (Comte, 1855; Cole, 1983; Fanelli, 2010; Simonton, 2006). We propose two ways in which the hydrological modeling network can increase the building of bridges with critical social sciences: firstly, through education, which will instigate structural changes in the long-term, and, secondly, through structural changes that can have an immediate effect.

First, education can facilitate the knowledge building necessary to understand the basic critical social science concepts (Arg. 7). Understanding basics of other scientific disciplines can increase communication and effectiveness in future work situations, enhancing inter-disciplinary collaborations (from Arg. 6). This teaching of social processes and reflexivity needs to be practical and integrated within hydrological modeling education (Micheletti et al., 2024; Oldfield, 2022; Stefanidou et al., 2014). For example, hydrological modeling education should have reflexivity and responsible modeling integrated in its curriculum: during a modeling course, the students learn to apply reflexivity as they model. Education should extend to working professionals in order to have them keep up with new insights and to also incorporate this knowledge in the current workforce.

Second, although education can help raise a new generation of hydrological modelers, we need **structural changes in the scientific network** to facilitate the incorporation of social aspects in daily modeling practices

(Arg. 8). Structural changes can guide and force the hydrological modeling network to adapt practices focusing on taking the social aspects into account (Jakeman et al., 2024). For example, funding requirements can include a positionality statement within the funding application (from Arg. 5) or a research plan that specifically designates time for active reflexivity. Also, journal requirements can be adapted to incorporate social aspects in hydrological modeling more explicitly. Journals might start asking for a positionality statement as well, or they can ask for documentation on assumptions in the modeling process.

# 315 6 Reflecting on what the hydrological modeling network can learn

Building a bridge to critical social sciences can improve transparency about the social aspects of hydrological modeling. Also, considering and disclosing the uncertainties associated with these aspects potentially creates more reproducibility. Increased transparency and reproducibility can contribute to more constructive scientific progress and more responsible and accountable policy making.

Also, acknowledging social aspects in hydrological modeling can open **new avenues for research** (Arg. 9). Critical social science understanding can move the hydrological modeling network towards more productively working on societal problems (from Arg. 7). Through reflecting, modelers are incentivized to rethink their modeling decisions. This might result in more robust, inclusive and accountable modeling decisions. In turn, this will provide more accountable decision-support. Reflexivity highlights assumptions made. Sharing these assumptions can streamline research where researchers can consciously build on each others methods or findings (Laplane et al., 2019). It is easier to know what has or has not been done before and to have the ability to complement each other because of that knowledge. Additionally, it could be that new research will specifically look for diversity, instead of a universal model (Baldissera Pacchetti et al., 2024; Horton et al., 2022; Savenije, 2009). Different researchers would facilitate diversity in approaches and therefore give a more complete picture of how the world is understood (Baldissera Pacchetti et al., 2024). This diversity can encompass the different contexts in which the modeling is shaped or in which the modeling is used.

With more transparency on the social aspects of hydrological modeling, modelers and also funders, commissioners and decision makers can **take responsibility for model results** (Arg. 10). This should be a shared responsibility, not just the modeler's. The interplay between these actors can create dynamics influencing the modeling. This interplay should be made more visible (from Arg. 5). Structural changes in the modeling net-

work (from Arg. 8) can facilitate this. Due to the transparency, modeling results will be more retraceable, and the limitations of a modeling study are more evident for and between different actors in the hydrological modeling network. Reflexivity on ontology can help modelers in their ability to recognize how their model results are partial, and might have looked different with another ontology. The transparency on the interplay influencing the modeling can provide better information for decision/policy makers, contributing to their ability to justify their policy decisions.

For instance, after flooding in Brisbane and surrounding, the model results based on which the dam was operated were questioned, and the organisations behind them were held accountable (Supreme Court of New South Wales, 2021). This example shows that the organisations using and providing model results need 345 to be able to take responsibility of them. Sharing responsibilities can take many forms, but it starts with curiosity for and openness to knowing, understanding and taking action on the social aspects of hydrological modeling. Another example, outside of hydrology, is that the modelers that simulated the nitrogen emissions for a newly planned airport in the Netherlands were investigated by the Public Prosecution Service, because there were clear indications that all modeling decisions were made such that the nitrogen emission was as low as possible (Adecs Airinfra Consultants, 2021; NOS Nieuws, 2022). Not surprising perhaps, if the executing company sells themselves as "aviation lovers", but also the result of a commissioner that has certain interests. As such it is a clear example of how modelers can be held accountable for their model results, while they also face forces from, for instance, funders.

# 7 Invitation to start acting

As potential follow-up actions, we suggest:

- If you are a model user (i.e. someone who analyses and uses model results), you can consider asking the modeler for their assumptions, and the trust that the modeler has in the model results. One way to explore this, together with the modelers, could be a serious game, such as "Adventures in Model Land" (Skinner et al., 2024).
- - If you are a modeler, you can consider to start reflecting on your positionality, and consider to include a positionaly statement in your next modeling study. How did your experience and position in society influence how you approached this study? Alba et al. (2025b) also recommend exploring

the auto-ethnographic approach proposed by Eitzel (2023). ter Horst (2025) proposes the value-ring method, which includes guiding questions about the potential influence of the model's application, and encourages adjustments to the model and modeling process when necessary.

- If you are teaching the next generation of hydrological modelers, you can consider incorporating reflexivity practices and social science basics in your lecture, computer practical, course, or curriculum. Somogyvári et al. (2025) offer an insightful example of how these elements can be incorporated into higher education. Additionally, the five points outlined in the manifesto by Saltelli et al. (2020) can serve as a guide for course development.
- If you are a commissioner, consider allocating additional time and funding within projects or even making it a formal requirement to support reflexivity in the modeling process. This could include requiring a positionality statement and the development of thorough documentation to enhance transparency, or organizing a focus group to examine the potential influence of the model, with subsequent adjustments to the model and modeling process if unintended effects are identified, as suggested by ter Horst (2025).
- If you are overseeing a modeling team, you can consider having a discussion on internalised assumptions in your way of working, also known as entrenched workflows (Levine and Wilson, 2013). Situated modeling, as suggested by Klein et al. (2024), could be a good starting point.
- In general, the "Translating Transformations" project developed a toolbox to enhance critical social science literacy. See www.translating-transformations.org/toolbox

These follow-up actions sound like a recipe. However, in this whole opinion paper, we have advocated and shown that hydrological modeling is context dependent. Therefore, we acknowledge that anyone implementing these potential actions needs to navigate their own working environment. More importantly, this list is not definitive; we invite you to explore and discuss this topic further, and come up with your own ways of incorporating reflexivity.

### 8 Conclusion

In this opinion paper, we argue why and how we think the hydrological modeling network, which we define as all actors, i.e. funders, commissioner, modelers, users, decision-makers, involved in and influencing the modeling study, can benefit from insights and practices from the critical social sciences. To support this, we have four pillars of arguments: the social aspects in hydrological modeling, insights from critical social sciences, building bridges between sciences, and reflecting on what the hydrological modeling network can learn. Based on these arguments, we provide some tangible follow-up actions targeting the whole modeling network to promote responsible modeling – modeling that is accountable, transparent, inclusive and reproducible, modeling that is is aware of the visions that were included and that were sidelined, and the ethical implications of representation. This responsibility is carried by all actors related to the modeling study. Even though we focused on the hydrological modeling network, we believe these lessons are also applicable to other modeling communities.

The main take-away, from our perspective, is that responsible modeling is a shared responsibility. We realize that modelers tend to already bear a lot of the responsibility and are the easiest ones to ask actions from. Substantial change is not possible without also addressing the other actors in modeling studies, such as educators, commissioners, funders or supervisors. Therefore, we address the complete modeling network and society. We invite all actors to take up their share in establishing responsible modeling.

Author contributions. Conceptualisation: JR, in consultation with LM. Visualisation: UP and JR. Discussions: all au-405 thors. Drafting of manuscript: JR with revisions from all authors. Revisions: LM, in consultation with all authors.

*Competing interests.* One of the co-authors is on the editorial board of the journal Hydrology and Earth System Sciences.

Acknowledgements. We would like to thank Derek Karssenberg and an anonymous reviewer for their constructive feedback which greatly improved our argumentation.

LM received financial support from the Dutch Research Council through a personal Veni grant (nr. 17297, entitled What about the modeler? The human-factor in constructing Earth and environmental predictions). UP was funded by the Swiss National Science foundation under grant number 217899.

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
