# Peer review of "HESS Opinions: Reflecting and acting on the social aspects of modeling"

_EGUsphere, 2025_

## Author Response (AR1)

**Reviewer 1**

**General Comments:**

Remmers et al. provide an opinion piece on how hydrological modeling could benefit from insights and practices of the critical social sciences. They offer a well-structured and argued discussion on reasons for and possibilities to increase the accountability, transparency and responsibility in hydrological modelling.

I particularly enjoyed the last part of the paper which has a lot of important and well communicated conclusions and action items for the different actors in a hydrological modelling network. I do believe that the introduction and motivational part of the paper can be strengthened by making new terms and ways of thinking more approachable to the reader inexperienced with critical social sciences. I therefore have a few suggestions that I hope can help in this regard.

Generally, I believe this is a very fitting contribution for HESS and a nice piece for the hydrological modelling community to reflect on current and future modelling practices and the impact social aspects might have on our work. Awareness is the first step to change, which is why I recommend publication after minor revisions.

Thank you for your encouraging evaluation of our work, and the constructive feedback provided.

**Specific Comments:**

[Are models perceived as neutral and objective tools?]

The authors base their motivation on the framing that models are perceived as neutral and objective tools. I would argue that most hydrologic literature (and also the sources cited in line 16) argue that models are hypothesis and therefore not quite as neutral as implied. These (model) hypotheses are generally formed based on a perceptual model which is then translated into the mathematical model that becomes the "tool" we use. As perceptual models are known to be personal and at least in part qualitative, I think we can agree, that by the time the model is formed and ready to be used as a "tool" a lot of social influence has already happened. The authors themselves describe part of this process in their Argument 2. Therefore, I keep on stumbling over the sentence "models are perceived as neutral and objective tools" as something I can't fully agree with. And I would imagine that this will be the case for most experienced modelers. To engage both groups (the problem aware and less aware modelers and model users) equally well, it might be helpful to simply acknowledge that different groups in the hydrological modelling network are more or less likely to see a hydrological model as a "neutral and objective tool", but that it is important for everyone to understand what this notion may lead to.

I believe that most of my discomfort comes from the sentence "Within hydrological modelling, a persistent notion exists that a model is a neutral, objective tool" that is used prominently in abstract and introduction. To me it has the disadvantage of veiling and softening the main

motivation for this paper (the assumption of a neutral and objective tool is questionable and comes with consequences) and giving an impression of consensus where a spectrum of understanding already exists.

I assume that this comes down to mere nuances of formulation as I realize that "notion" is supposed to imply that "many believe models are objective, but this view is not universally accepted". I argue, however, that a more direct phrasing of this issue will help the reader to grasp the main point and motivation of this paper more easily and helps to acknowledge that we do not start at zero regarding the awareness of this problem.

I therefore suggest to either change the first sentence of the introduction to be a more direct description of the problem or include a short discussion of the different states of awareness regarding this problem around line 24. I believe this would also make the citations from line 16 more fitting (see minor comments).

We can fully resonate with the feedback from the reviewer on our sentence that "models are generally perceived as neutral and objective tools". Among many modellers there is wider acknowledgment about the expertise and subjectivity involved in developing the perceptual model, and also the lack of data that leads to methodological underdetermination is generally well recognized – usually in terms of uncertainty. We do feel, however, that this awareness is not always linked to notions of bias, power, or non-neutrality. Furthermore, commissioners and model end-users are not always aware of subjectivity and uncertainty involved with models. To reflect this better, we have reformulated the abstract and the introduction paragraphs.

[What is critical social science and how can we benefit from it?]

As a reader I am very interested in what critical social science is and how we can benefit from it. But from the introduction alone I feel I do not yet see what critical social science has to offer that hydrology can learn from. I feel that might mainly be the case because the introduction could often benefit from some specific examples that guide and convince the reader of the storyline instead of making statements that are justified with citations from a field less familiar to the average hydrology reader. I would prefer to be convinced through examples from the literature rather than expected to read all the cited papers myself to reach a similar conclusion. I would appreciate if the authors could include more specific examples from the papers they cite when building their argument in the introduction. More details and general contemplations are then provided in the following chapters. I will provide specifics in the minor comments.

We thank the reviewer for this suggestion, for it is of course essential to keep readers on board and therefore to be clear on what we mean with the critical social sciences. We have now included two examples, one from STS and one from political ecology, that demonstrate the kind of analyses done in this field. Subsequently, we included a paragraph that provides examples specifically for hydrological modelling, that shows how this can be relevant.

**Minor/Technical Comments:**

Abstract – "marginalizing certain stakeholders": is this the most relatable problem to
mention at this point? I initially fail to imagine an example of what this might mean and
would like to read an "OR" with a more relatable example (maybe overconfidence in
model results etc.) or a more specific example of the marginalized stakeholder
consequence.

Yes, for us marginalized stakeholders is key – "overconfidence in model results", as suggested by the reviewer, can also lead to certain stakeholders being marginalized; some people or entities pay the price for this overconfidence. Staying close to the modelling, marginalized stakeholders are those voices that are not heard or represented in the modelling. In a decision-support context, this effect is clear: if models only evaluate the effect on discharge and not on fish population, decisions focused on discharge might negatively affect not only the fish population but also communities depending on these fish. In a scientific context the effect is less direct, but model results can shape discourses, foreclosing alternative frames. One of these discourses, one could argue, is the focus on discharge in hydrological modelling, while for many questions perhaps other fluxes or states are more relevant. For instance, in the Netherlands flood risk mitigation is always prioritized above optimizing biodiversity.

Given that this is the abstract we have added a short example: "However, this notion has several, potentially harmful, consequences. One is the marginalization of certain stakeholders: failing to acknowledge or incorporate alternative perspectives on the issue, which might have warranted a different (modelling) approach."

Abstract – "The main take-away, from our perspective, is that responsible modelling is a shared responsibility" – This sentence might diminish the contribution of the article a little. I would suggest rephrasing in a way that lists the different contributions of the article. E.g.: We highlight that responsible modelling is a shared task between all actors of a modelling network and provide several actionable recommendations for individual actors to increase their share in facilitating responsible modelling. Or something similar.

We agree and have incorporated the suggestion from the reviewer.

• L25-26 – I believe these citations see models not as a neutral tool but as a hypothesis that needs testing. I therefore find the referencing questionable with the current phrasing. Especially, since the same citations are used in line 27 when stating that "models are simplifications where we need to make choices on what to represent or not to represent". Please refine citation usage for these two sections of the paper.

These papers were cited in these contexts because they also make the claim that models are generally perceived as neutral – to subsequently attack this claim. Given that we have rephrased this sentence in response to the first point raised by this reviewer, we feel we could make the newly proposed sentence without any further citation.

L21-23 – I think this part would benefit from at least one very specific example. I can offer
a potential example of first nations in Canada suffering from not being included as
stakeholder during dam construction. Maybe the introduction of this paper can be a
good starting point to investigate specific examples:

https://www.tandfonline.com/doi/full/10.1080/08941920.2018.1451582

Thank you for this suggestion. Dam construction is indeed exactly an example where models are used to justify the construction, while many examples exist where certain stakeholders are not involved in the process and as such, marginalized. We have included two other examples to clarify our case, that align with the examples that we brought in earlier for the critical social sciences. We have decided to move these examples to after the introduction of the critical social sciences (with the examples from STS and political ecology).

"Examples of how certain perspectives might be prioritized in model development are provided by Packett et al. (2020) along the lines of gender. They cite a case studied by Zwarteveen (2017) in Nepal, where men and women worked cooperatively as co-farmers but prioritized different aspects of water flow. Men, responsible for land preparation, focused on water arriving at the start of the irrigation season, while women, who managed weeds, needed consistent water throughout the season. An irrigation distribution model optimized for either water arrival or water sustainment would thus benefit either men or women in their activities. Nabavi (2025) presents a case that illustrates the broader socio-political context of modelling. In this instance, a hydrological model was employed to justify an interbasin water transfer to the historically significant city of Isfahan, Iran. The transfer was underpinned by a century-old narrative, with the model serving primarily to reinforce this story, framing upstream water as "lost" to the Persian Gulf unless redirected to Isfahan. In response, upstream communities developed a counter-model that accounted for ecological impacts and the livelihoods of upstream populations. Within this alternative framing which also emphasized upstream effects, the justification for the water transfer no longer held."

• L28-29 – "This can result in injustices: some groups being overlooked [...]": I find it very difficult to jump between processes that become invisible vs. groups being overlooked etc. These are very different aspects of modelling consequences, and I believe it would be helpful to elaborate a bit on potential path dependencies or describe these different aspects in a bit more context than currently done.

We hope that the examples we have suggested above make the jump somewhat smaller.

- L35 the comma should be a dash to fit the beginning of the sentence?
- Agree, we adapted this.
- L38 STS as an abbreviation that is not used again in this paper, so it can be removed

We refer to STS more often now that we included an example.

• L38 – "provide insights into how to analyze and deal with non-neutrality": Would it be helpful to include an example of what is being done in this science so the hydrological reader gets and idea what might be worth implementing? This might provide further support to the next sentence calling for more responsible modelling.

In response to the suggestion above, we propose to include an example from political ecology and STS on how these fields contribute to understanding and dealing with non-neutrality. Together with the hydrological modelling examples, we think the introduction now more convincing and compelling (for which we would like to thank the reviewer!).

• L80 ff – "Proske et al." and equifinality in cloud microphysics. I would argue we have good examples of equifinal model performance in hydrology. I would suggest using a hydrology example here?

Yes, true. We have adapted the text to a hydrological example based on:

Khatami, S., Peel, M. C., Peterson, T. J., & Western, A. W. (2019). Equifinality and flux mapping: A new approach to model evaluation and process representation under uncertainty. Water Resources Research, 55, 8922–8941. https://doi.org/10.1029/2018WR023750

• L104 – something seems to be wrong with the citation (?), please check

Thank you, this has been fixed. The following citation was missing:

Mark S. Reed, Anil Graves, Norman Dandy, Helena Posthumus, Klaus Hubacek, Joe Morris, Christina Prell, Claire H. Quinn, Lindsay C. Stringer, Who's in and why? A typology of stakeholder analysis methods for natural resource management, Journal of Environmental Management, Volume 90, Issue 5, 2009, <a href="https://doi.org/10.1016/j.jenvman.2009.01.001">https://doi.org/10.1016/j.jenvman.2009.01.001</a>.

• L106 – consider removing the "obviously"

**Agree.**

• L124 – do the critical social sciences or a specific publication provide some sort of glossary or terminology framework that could be referred to here? If a hydrologist would want to learn about this vocabulary, where could he start?

We have closed this paragraph now with a sentence where we refer to Moon and Blackman (2014) and Wesselink et al. (2017).

Moon K, Blackman D. A guide to understanding social science research for natural scientists. Conserv Biol. 2014 Oct;28(5):1167-77. doi: 10.1111/cobi.12326. Epub 2014 Jun 24. PMID: 24962114.

• L144 – is there any example or guide on how to start if an author would want to write and add a reflexivity statement to their work?

We have included a reference at the end of this paragraph to the following papers;

Kirsti Malterud, Qualitative research: standards, challenges, and guidelines, The Lancet, Volume 358, Issue 9280, 2001, Pages 483-488,https://doi.org/10.1016/S0140-6736(01)05627-6.

Holmes, Andrew Gary Darwin. "Researcher Positionality - A Consideration of Its Influence and Place in Qualitative Research - A New Researcher Guide." Shanlax International Journal of Education, vol. 8, no. 4, 2020, pp. 1-10.

• L157-158 – "can have ethical implications in society AND water management"?

Agree, we adapted this.

• L160 – Is there one outcome for the development of ethics of artificial intelligence that could be named as being useful/adaptable to hydrology?

Yes, we agree that a more concrete example could be useful here. Many of the development of ethics in AI and responsible AI is useful and adaptable to hydrology, we provide one concrete example of a framework that could fit hydrological numerical modelling, based on Nabavi and Browne (2023).

Nabavi, E., Browne, C. Leverage zones in Responsible AI: towards a systems thinking conceptualization. Humanit Soc Sci Commun 10, 82 (2023). https://doi.org/10.1057/s41599-023-01579-0

 L165 – and again it would be great to read an example to make these new abstract ideas easier to grasp

Here an example on how ontology shapes hydrology.

"For example, hydrologists often distinguish between epistemic and aleatoric uncertainty. Recognizing aleatoric uncertainty, that is, uncertainty due to inherent randomness in natural processes, presupposes a belief that the world is not entirely deterministic. This illustrates how one's worldview, or ontology, influences which types of uncertainty are considered meaningful to study. The same applies to epistemology, the theory of how we know what we know. Modeling aligns well with a Newtonian perspective, which assumes that natural laws can be discovered and represented objectively. In contrast, a constructivist would argue that all knowledge is socially constructed, and thus would immediately question the idea of a single 'best' model, highlighting the partial and situated nature of modeling."

• Title for 4 – just a personal preference, but I would probably write "building bridges between (two) scientific disciplines" – but up to the authors

We agree with the suggestion as it better reflects the section, and will therefore adopt it. Thanks for the suggestion!

• I really like part 4! Do you have suggestions on how teachers should be educated/ can educate themselves on this if they would like to incorporate it in their classes? I asked this before, but can you maybe reference sources that would help the motivated reader to get started on writing a positionality statement?

We will again refer to the positionality reference mentioned above as a starting point. Furthermore, in response to reviewer 2, we will include our own positionality statement. In response to the question on Section 6 below, we added more concrete examples there.

L208 – The sentence about flexible modelling frameworks seems a bit detached. Or at least the context of why it comes up here does not seem to be explained in a convincing way. Maybe the authors can consider rephrasing the sentence and making the connection between diversity of approaches, flexible modelling frameworks and different context a bit clearer.

We agree that this sentence was out of place here, we have removed it.

• L235 – should there be a period/full stop at the end of the sentence?

Yes, thank, this is added.

• Section 6 – is there a possibility of providing an example for each point mentioned to make it easier for the reader to find a starting point? E.g. what type of assumptions could a model user ask for that might be relevant. How does he know what to ask for? Is there an example of a positionality statement a modeler could look at? Are there resources for reflexivity practices? Are there resources available each actor could look at to get started? To avoid people taking this as recipe you already have the follow up statement that anyone needs to adapt all this to his own working environment.

We have added concrete suggestions for each of the points mentioned. Thanks for this suggestion! ]

Conclusion – it might be helpful to have the definition of what you consider a hydrological modelling network to be a bit earlier then in the conclusions.

Agree. The definition is now added to the first time that we mention this term, in the introduction.

 References – ter Horst et al. "Making a case for power-sensitive water modelling: a literature review" is still cited as a discussion paper. But the final version of the paper is already available: <a href="https://hess.copernicus.org/articles/28/4157/2024/">https://hess.copernicus.org/articles/28/4157/2024/</a>

We have adapted it.

**Reviewer 2**

This is an interesting opinion paper providing an overview of the current research on social aspects of modelling, and proposing actionable recommendations for the modelling community. Please find my suggestions for further improvement below.

**Figure 1**

In my opinion the figure is not very useful in this form. For some arguments it is not obvious where they should be positioned (or it is obvious in which case the figure is arguably not very useful). I am also wondering why 'Model Problem' is on the left side (where 'Society' is) while the 'Modeller' is on the right side (where 'Modelling Community' is). It could also be the other way around. However, I do not have a suggestion for improvement so you could also keep it, possibly with minor adjustments. An alternative would be to organize the figure along the lines of the text, i.e. social aspects, insights from social sciences, building bridges between sciences, reflecting (note that this goes, more or less, from defining the problem (could be left in the figure) towards possible solutions (could be on the right side of the figure)).

Thank you for the suggestion to rethink the figure. It is true that "Model problem" is placed in "Society", but note that the "Modeller"-circle and "Modelling community" -circle also overlap with society. We will emphasize these overlaps in more detail, because they are the core of our story: modellers, and the modelling community, are part of a society, and address problems that are embedded in society. We will explore different configurations of the figure to see how we can improve clarity. We want to emphasize the overlap, because that is what this figure aimed to convey.

**Line 15**

'hydrological modelling'. Please define, what type of models? For instance, is the discussion here about model concepts/equations or (also) about the software implementation? Also, is the discussion about forward simulation models (any form) or also about models relying on statistical learning (including machine learning) that are mostly not run forward in time – note that in both models types, observational data are used and there are currently blends, often

referred to as hybrid models. Also, do you refer to the activity of model building? Or also other steps in the model development cycle (e.g. calibration, application).

Good point. We mean: The practice of hydrological modelling, from developing and implementing model code, to applying the model (including for instance calibration) to address a certain question or issue. We have now clarified this right in the first paragraph of the paper.

**Line 15-18**

It is argued here that models are neutral because they are influenced by society. This is indeed the case. However, in addition, models are influenced by the social network within the modelling community (see e.g. Babel et al, 2019). Thus, social factors within the modelling community as well as influences from outside (society) are important in making models non-neutral. In my opinion both aspects need to be highlighted here.

In addition, it would be good to define 'neutral'. It seems you consider it as a synonym for 'objective' but these may be different concepts. It seems the references provided do not clearly define 'neutral'.

Thank you for the suggestion. Objective and neutral are much related: if a model is regarded as objective (a true representation of the reality) it is also seen as neutral (not biased towards one particular interest). However, a model is always partial, so not objective, and therefore not neutral.

Yes, the modelling community also shapes the way we model, this is our "Argument 2" (The modelling process itself is a social product). We have added this to the introduction ("Models are shaped and influenced by social and political dynamics (both at societal level and within the modeling community) and, in turn, influence them")

We have now also added how we understand neutrality, namely as "remaining impartial or not taking sides".

**Line 30-31**

'Simultaneously, ignoring the political side ...' I have a similar comment here as given above (line 15-18). This sentence (line 30-31) seems to imply you consider mainly influence from society ('political side of models') on the model (and model community). However also within the model community social factors influence modelling.

Agree, in response to reviewer 1 we have included explicit examples here. In response to this comment, we also included a power-example from within the scientific community.

Section 2 Social aspects in hydrological modelling

You distinguish three 'arguments'. The description of these need to be improved in my opinion. Please let me explain how I see it and how I recommend describing this. You are free following a different approach but please consider my line of reasoning below.

The first argument (in my opinion) should be about how society affects models and modelling. Society is a stakeholder and as a result, society influences the 'shape' of models. The current text however only explains \_that\_ models are embedded in society. In addition (and more importantly), it needs to describe that society influences models/modelling and how.

We understand the position of the reviewer and mostly agree (although we argue that society not only influences models because it is a stakeholder, but also because modelers are part of society and therefore societal ideas get incorporated in the model). Right now we only describe how the problems we address with models are embedded within society, but not that these models are a product of this same society (and argument we made in Melsen et al. 2018). We added the following sentence at the end of arg 1:

"That being said, it should be recognized that not only the challenges addressed with models are embedded in society, but that the modeling itself is also the result of the society in which it was shaped (Melsen et al., 2018, Riaux et al. 2023). Norms, values, and discourses commonly accepted with a society provide the space within which the hydrological model is developed and accepted. Even more, what is considered a problem is determined by societal standards held by the model commissioners, modelers and model-users. For instance, flood risk might be considered differently at different places."

Similar to the first argument, the second argument needs to describe that social aspects within the modelling community influence models/modelling and how. It does. However, in my opinion it can be improved by also explaining the mechanisms. Equifinality is relevant here (multiple models are 'possible') but the mechanisms that lead to these particular different models are equally important and could be described. One of the mechanisms is 'habits', as described in Babel et al (2019). You use it as a reference for equifinality but if I am correct, we did not discuss it in our paper (Babel et al (2019)). The paper mainly explains \_how\_ social factors lead to particular models (what you would call 'non-neutral' models).

We understand and agree with the reviewer. The reference to Babel et al. (2019) was not intended to refer to the equifinality but to the social processes in that sentence, but we have rephrased this part to make the examples more concrete.

The third argument, in my opinion, should be about the fact that models have implications for society (political, ethical) and that it is thus extremely important (also outside academia) to describe and discuss how 'neutral' they are as they have impact outside academia. This third argument does not discuss how social factors influence models (like the first and second argument). Instead, it describes the relevance of modelling choices for society. I do not see how 'the previous arguments come together' (line 98) here (at least for me it is a confusing statement); if you are convinced this is the case please improve the explanation.

We have removed the statement about arguments coming together (we meant that both arguments contribute to the political and ethical implications of models). We have rephrased the beginning of the paragraph, which we think indeed contributed to a better flow of the story.

**Line 160**

Ethics from AI could indeed be used for ethics in numerical modelling (is this manuscript about numerical modelling only?). In my opinion this deserves somewhat more discussion. For instance, try to summarize the ethics field in AI and give suggestions how it could be converted to numerical modelling (or what we could learn from it).

Thank you for this relevant question – the same was brought forward by reviewer 1. In response, we have now included the description of one of the frameworks that was developed for

responsible AI, and discuss how this framework can in principle be 1:1 copied to numerical modelling.

Nabavi, E., Browne, C. Leverage zones in Responsible AI: towards a systems thinking conceptualization. Humanit Soc Sci Commun 10, 82 (2023). https://doi.org/10.1057/s41599-023-01579-0

**Positional statement**

Consider including a positional statement (or a short description of the background of the authors).

This is indeed a very good point and the least we could do. We have now included a positionality statement.

Minor comments

Line 20

Reference(s) seem to be missing (after Packett et al, 2020)

This is fixed.

Line 22

'might'. Consider 'may' or 'will'

We have reformulated several parts of this section, and now use "may".

Line 64

'purely technical'. Technical does not need to be neutral (not at all, see e.g. work by Latour). Reword and avoid 'technical' here.

Agree, we have reformulated this sentence.

Line 110

'tools and theoretical frameworks', rewrite 'theoretical frameworks and tools' (theory comes first, tools are derived from the theory).

Agree, we have switched the order.

---

## Author Response (AR2)

**Rebuttal**

"HESS Opinions: Reflecting and acting on the social aspects of modeling"

We would like to thank the reviewers for their encouraging response, and for their help with settling the final details. Below we shortly respond to the minor issues raised. Besides, we have included one more suggestion in Section 7, as we recently came across a toolbox to stimulate critical social sciences literacy. This is added as a last point to the section.

**Reviewer 1**

The revision has resulted in a considerable improvement of the manuscript. I do not have additional comments apart from a number of suggestions to further improve Figure 1.

The left side refers to 'Model results'. This is somewhat unbalanced as it is a model 'output' while the 'Modeller' on the right side is a model 'input'. My suggestion is to replace 'Model results' by 'Stakeholders' (specific persons from the Society that are linked to a particular model), influencing model building just like the 'Modeller' on the right side influences model building (specific persons from the Modelling community linked to a particular model).

Also, consider adding an arrow in the orange part from 'Model' to 'Modeling community' (just like the smaller arrow 'Models have societal and ethical consequences' but from left to right) referring to the feedback of models to the modeling community. This feedback involves model outputs (from which the community improves/changes understanding), model concepts (advancement of understanding of processes within the community) as well as model software (the modelling community tends to stick to tools (and thus concepts) that have been used by others). This line of reasoning is also described in Section 3 (Arg. 1).

The figure could be connected more explicitly to the main text. The blue circles refer to Arg. 1 (Section 3), the orange ones to Arg. 2 (Section 3), while the smaller pointer at the bottom ('Models have societal...') refers to Arg. 3. The argument numbers could be given in the figure or in the caption while in Section 3 please consider referring to the Figure.

We have incorporated the majority of the suggestions. We agree that 'model output' was not in balance with 'modeler', and have reformulated it, as suggested by the reviewer, to 'stakeholders'. We tested including an arrow from 'model' back to 'modeling community' – as we agree with the reasoning of the reviewer that this feedback loop exists - but realized that this required quite some extra explanation in the text. Therefore, we have decided to leave it out in the final figure. We have included a more elaborate caption, referring to the arguments, and have included more references to the figure in the main text.

**Reviewer 2**

I congratulate the authors on their revised manuscript. These are great improvements, and I found the story line very convincing and well-argued. Their points are now well supported with examples and literature, and I believe this comment will be an important contribution for the modelling community. I hope it will be read extensively and thank the authors for their thorough revisions.

I only noticed a few minor typos that could be changed before publication.

L145ff – the "already present in and around hydrological modeling" is very similar to the next sentences "already embedded in hydrological modeling can". This feels a bit doubled. Maybe the sentence can be combined.

L256 – the positionality is in section 2 not 3

L364 – missing space after ter Horst (2025)

We would like to thank the reviewer for these suggestions, which have all been incorporated in the text of the manuscript.

On behalf of all co-authors, Lieke Melsen